# Lithium systematics in global arc magmas and the importance of crustal thickening for lithium enrichment

Chen Chen [1,2,3], Cin-Ty A. Lee [1,4 ✉], Ming Tang[1,5], Kevin Biddle[1,4] & Weidong Sun [3,6,7,8]

Much of the world's Li deposits occurs as basinal brines in magmatic orogens, particularly in continental volcanic arcs. However, the exact origin of Li enrichment in arc magmatic systems is not clear. Here, we show that, globally, primitive arc magmas have Li contents and Li/Y ratios similar to mid-ocean ridge basalts, indicating that the subducting slab has limited contribution to Li enrichment in arc magmas. Instead, we find that Li enrichment is enhanced by lower degrees of sub-arc mantle melting and higher extents of intracrustal differentiation. These enrichment effects are favored in arcs with thick crust, which explains why magmatism and differentiation in continental arcs, like the Andes, reach greater Li contents than their island arc counterparts. Weathering of these enriched source rocks mobilizes and transports such Li into the hydrologic system, ultimately developing Li brines with the combination of arid climate and the presence of landlocked extensional basins in thickened orogenic settings.

[1] Department of Earth, Environmental and Planetary Sciences, Rice University, Houston, TX 77005, USA. [2] CAS Key Laboratory of Mineralogy and Metallogeny, Guangzhou Institute of Geochemistry, Chinese Academy of Sciences, 510640 Guangzhou, China. [3] University of Chinese Academy of Sciences, 100049 Beijing, China. [4] Center for Energy Studies, Baker Institute of Public Policy, Rice University, Houston, TX 77005, USA. [5] School of Earth and Space Sciences, Peking University, 100871 Beijing, China. [6] Center of Deep-Sea Research, Institute of Oceanology, Chinese Academy of Sciences, 266071 Qingdao, China. [7] Laboratory for Marine Mineral Resources, Qingdao National Laboratory for Marine Science and Technology, 266237 Qingdao, China. [8] Center for Ocean Mega-Science, Chinese Academy of Sciences, 266071 Qingdao, China. ✉email: ctlee@rice.edu

Growth in the hybrid and electric vehicle industry is placing increasing demand on lithium (Li) needed for rechargeable batteries[1–4], motivating research on the origins of Li deposits. Most of Earth's economic deposits of Li come from hard rocks (pegmatites) and basinal brines, the former representing extremely fractionated magmas or magmatic fluids and the latter representing Li-bearing groundwaters in evaporative basins (Fig. 1a, b; see Supplementary Dataset 1 for details). Smaller numbers of Li deposits may be found in clays, oilfield brines, and byproducts of geothermal wells[1,4]. Of interest here is the geologic distribution of basinal brines, which primarily occur in continental arc subduction zones or collisional settings, such as the

**Fig. 1 Distribution of global lithium deposits and conceptual cartoon describing Li cycling in subduction zones. a** Map of continental brine, clay, pegmatite (hard rock), geothermal, and oilfield brine deposits based on compilations in this study (see Supplementary Dataset 1). Continental Li brines are dominantly found in Cenozoic/Mesozoic orogenic belts; pegmatites are found in eroded magmatic orogens (maps based on Roberts and Bally[59]). Base map was processed with GeoMapApp (http://www.geomapapp.org)[60]. **b** Map of estimated Li resources with size of symbols reflecting ktons of Li metal. **c** Map of Li reserves with symbol size corresponding to ktons of Li metal. Resources represent the amount of available Li, whereas reserves represent that which can be extracted economically. Data are based on company reports compiled in this study (see Supplementary Dataset 1). **d** Conceptual model for Li cycling in subduction zones. Relative to background mid-ocean ridge basaltic volcanism, Li concentrations are enhanced in arc magmas by direct contributions from the subducting slab, melting in the mantle wedge, and fractional crystallization in the upper plate. Crystal fractionation is more extensive in thick continental arcs. In addition, mature continental arcs are often associated with net-evaporative intermontane basins. Weathering of andesitic and rhyolitic source rocks in thick continental arcs, followed by transport of solutes into these local basins, leads to further enrichment of Li. In island arcs, appropriate source rocks are scarce and any mobilized Li is likely lost to the ocean.

Andes and Tibetan plateau. Basinal brines are not as prevalent in all subduction zone settings, which is likely because the occurrence of Li brines is the result of the unique combination of ideal source rocks and the presence of basinal reservoirs for secondary enrichment in continental arc settings.

In the context of basinal brines, this paper focuses on the origins of Li-enriched arc magmas, which serve as the source rocks supplying soluble Li into landlocked basins. One widely held view is that arc magmas are enriched in Li due to contributions from slab fluids or melts[5–10]. However, separating slab contributions from the effects of endogenic processes, such as melting of the mantle wedge, intracrustal differentiation, or crustal contamination is challenging. Numerous attempts have been made with Li isotopes, but results and interpretations have not been conclusive or straightforward[5,6,11,12]. Thus, to improve our understanding of how Li becomes enriched in magmas in different types of arc settings, we examine here the Li elemental systematics of global arc magmas and evaluate the relative importance of the subducting slab, decompression melting of the mantle, and intracrustal differentiation in generating Li-enriched magmas.

## Results

**Li evolution trends in arc magma differentiation.** Lithium becomes enriched in arc magmas with differentiation (Fig. 2a, b) as can be seen by the positive correlation between Li and $SiO_2$. Such behavior is consistent with Li behaving as an incompatible element during differentiation[13–17]. The ratio of Li to K, a perfectly incompatible element during most of the differentiation process[18–21], remains relatively constant or slightly decreases with $SiO_2$ (Fig. 2c), suggesting that Li is highly to moderately incompatible and hence becomes concentrated in residual magmas with progressive crystal fractionation.

There are, however, differences in absolute Li concentrations between arcs of different crustal thickness. In Fig. 3a, we have plotted average Li concentrations of primitive magmas for individual arc segments against their characteristic crustal thicknesses. Excluding the Mariana island arc, which suffers from small sample size in terms of Li measurements for primitive magmas and extreme alteration of those samples (see Supplementary Fig. 1 for details), there is a positive correlation between the Li contents of primitive arc magmas and crustal thickness, with primitive magmas of thin arcs (<25 km) converging towards mid-ocean ridge basalts (MORB, 6.5 ppm[22]) and primitive magmas in thick arcs (>40 km) approaching ~8 ppm Li (Fig. 3a). These differences in the Li contents of parental magmas are inherited throughout subsequent intracrustal differentiation as can be seen by the systematically higher Li contents for arc magmas from thicker crust for a given $SiO_2$ content. Such inherited effects on the behavior of Li during differentiation can be replicated with simple crystal fractionation models using the

initial Li concentrations corresponding to magmatic series for a given crustal thicknesses (Fig. 2a, b).

**Li/Y systematics in arc magmas.** Li/Y systematics in arc magmas may shed additional light on subduction Li input. During mantle melting, Li is thought to behave like Y[5,6,9,23], a moderately incompatible element. Because Li is thought to be fluid mobile[7] and Y relatively fluid-immobile[24–26], the contribution of slab fluids or hydrous melts to the mantle wedge should lead to elevated Li/Y in primitive arc magmas[5,9]. However, our results show that the Li/Y ratios of primitive arc magmas are not as anomalously high as suggested by prior studies[5,7,10](Fig. 3b). It can be seen that Li/Y ratios in primitive magmas of thin arcs converge to that of MORB (~0.2[9,22]) but, like Li in primitive magmatic rocks, Li/Y of primitive magmas shows only a subtle increase with increasing crustal thickness. This subtle correlation may result from residual garnet in the mantle source, which sequesters Y, consistent with observed positive correlations between Dy/Yb of primitive arc magmas and crustal thickness[27,28] or Li/Y (see Supplementary Fig. 2 for details).

We find that Li/Y ratios of arc volcanics can fractionate depending on differentiation pressure (Fig. 2d). For arcs with crustal thicknesses less than ~25 km, Li/Y remains constant. For arcs with crustal thicknesses greater than ~30 km, Li/Y increases progressively with $SiO_2$, which we attribute to the appearance of amphibole and/or garnet as fractionating phases in thicker crust[29]. The compatibility of Y is increased in the presence of these minerals[30–32], resulting in depleted Y and increasing Li/Y with differentiation in thicker crust. The effect of amphibole and garnet in decoupling Li and Y is also corroborated by correlations between Dy/Yb and Li/Y in arc magmas (see Supplementary Fig. 3 for details). Previous studies suggested a slab origin based on high Li/Y in evolved arc magmas. These magmas, however, had already experienced Li/Y fractionation, generating an apparent slab signature (Fig. 3b).

**Li systematics versus slab thermal parameter Φ.** To further evaluate the role of slab fluids in Li enrichment of arc magmas, we also compare Li and Li/Y ratios of primitive arc magmas to the slab thermal parameter Φ (Fig. 3c, d), which is defined as $\Phi = tv\sin\theta$, where $t$ is the age of the slab, $v$ is the subduction velocity and $\theta$ is the slab dip[33]. Slab thermal parameter is a relative measure of the thermal "inertia" of the subducting slab at a given depth, and thus indicates the extent to which the slab may dehydrate at sub-arc depth. For example, a young, slowly descending slab would be expected to be hotter than an old, rapidly descending slab at the same depth, resulting in greater extent of dehydration and release of fluid-mobile elements from the young, hot slab. However, using the slab thermal parameters extracted from Syracuse et al.[34] (Supplementary Dataset 2), we find no correlation with primitive arc Li and Li/Y (Fig. 3c, d).

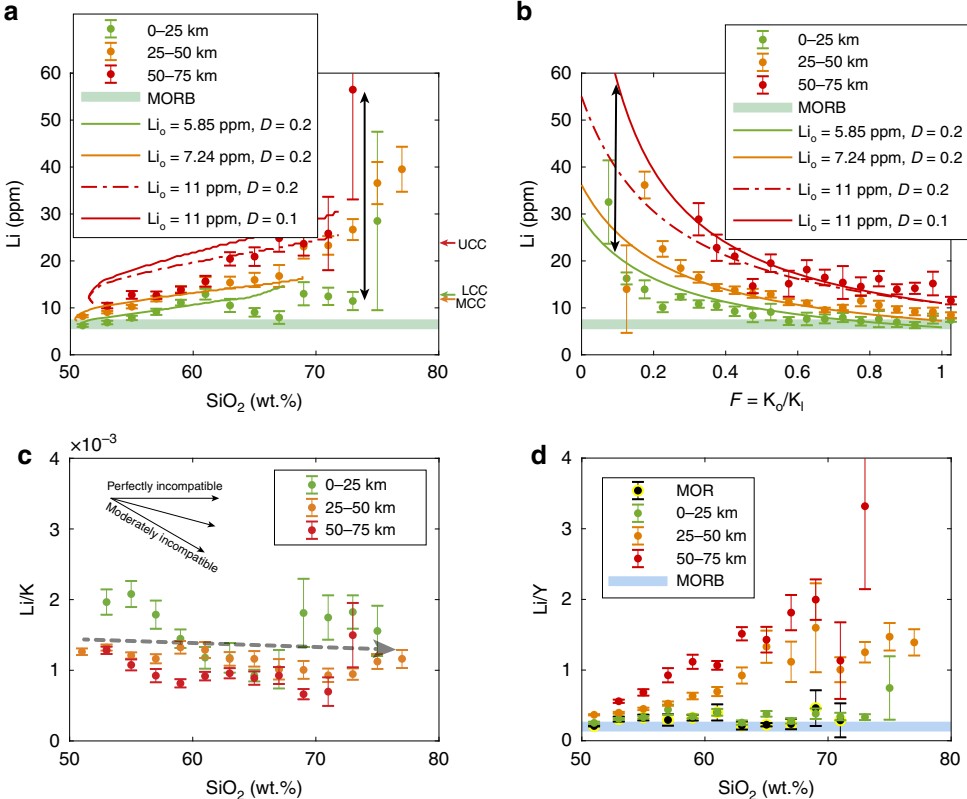

**Fig. 2 Evolution of Li systematics in arc magmas for arcs of different crustal thicknesses. a** Li (ppm) vs. SiO$_2$. **b** Li (ppm) vs. residual melt fraction ($F$) as determined by the relative enrichment of K. **c** Li (ppm)/K (ppm) vs. SiO$_2$. **d** Li (ppm)/Y (ppm) vs. SiO$_2$. Arc magmas were divided into three crustal thickness groups (0–25 km, 25–50 km, and 50–75 km) based on calculated crustal thickness (see "Methods"). For each crustal thickness interval, samples were averaged in intervals of 2 wt.% SiO$_2$ or 0.05 residual melt fraction. Error bars on Li, Li/K, and Li/Y represent two standard error (2se). In (**a**) and (**b**), solid and dotted lines show modeled residual melt compositions due to crystal fractionation for different initial Li content (Li$_o$) and bulk partition coefficient (D) (see "Methods"). Pale green bar in (**a**) and (**b**) denotes average Li concentration of mid-ocean ridge basalt (MORB) from Gale et al.[22]. Pale blue bar in (**d**) shows the Li/Y ratio of MORB from Ryan and Langmuir[9]. The colored arrows in (**a**) show average Li contents of upper, middle, and lower continental crust from Rudnick and Gao[61]. Arc data are extracted from GEOROC[56]. Mid-ocean ridge data are from Keller et al.[62].

## Discussion

The first question is how important slab contributions to Li in arc magmas are. Because sediments and altered oceanic crust are often enriched in Li compared with unaltered oceanic crust and because Li is soluble in aqueous fluids, it is widely thought that slab-derived fluids contribute significant amounts of Li to arc magmas[5–10,35–37]. Our observations, however, suggest that slab contributions to Li are probably not significant. As shown above, Li and Li/Y ratios of primitive arc magmas converge to those of MORB with decreasing arc crustal thickness. In addition, primitive Li and Li/Y ratios are not correlated with slab thermal parameter, which might be expected to correlate with the extent of slab dehydration. Our observations thus suggest that most of the Li in primitive arc magmas come from the mantle wedge with minimal amplification from the slab. Our results are consistent with the absence of slab-derived Li isotopic signatures in most arc magmas[11,12,38,39]. The lack of slab Li in arc magmas indicates that Li is not released from the slab or that the slab is not itself enriched in Li. Alternatively, if Li is released from the slab, such Li does not make it to the source regions of arc magmas[12].

We can also evaluate the importance of crustal thickness in controlling Li content of arc magmas. Correlations of average Li with crustal thickness, as shown in Figs. 2a, b, 3a, and 4b, suggest that Li enrichment in arc magmas is controlled by endogenic processes, that is, the combined effects of the extent of melting in the mantle wedge and the degree of crystallization during intracrustal differentiation. The fact that Li contents in primitive arc magmas are higher for thicker arcs is similar to that seen for other incompatible elements, which has been interpreted as the effect of thick crust on suppressing the extent of decompression melting in the mantle wedge, thus increasing incompatible element concentrations in the melts[27,28,40]. Based on our results (Fig. 3a), extrapolation to crustal thicknesses beyond 60 km, such as beneath the Andes today, suggests that primitive arc Li contents can be increased by up to a factor of 2 via crustal thickening.

After magmas leave their mantle source regions, intracrustal differentiation superimposes additional Li enrichment due to the incompatible behavior of Li. To first-order, the degree of differentiation is enhanced in thick arc crust, as evidenced by the andesitic composition of magmas in thick continental arcs compared to thin island arcs, which are basaltic[41–44]. This effect is most likely explained by a longer transcrustal transit time, which leads to more differentiation in magmas traversing thicker arc crust. The more evolved magmas in thick arcs are thus expected to be more enriched in silica and Li than thin arc magmas (Fig. 4a, b).

In summary, the combined effects of crustal thickening on suppressing mantle wedge melting and enhancing intracrustal differentiation lead to a 4-fold enrichment in the average Li content of magmas erupted through thick arcs compared to MORBs (Fig. 4b). In particular, highly evolved rhyolites erupted at thick arcs exhibit >10-fold enrichments relative to parental basalts, reaching Li contents as high as 50–100 ppm (Fig. 2a, b). The resulting Li-enriched andesites and rhyolites serve as ideal

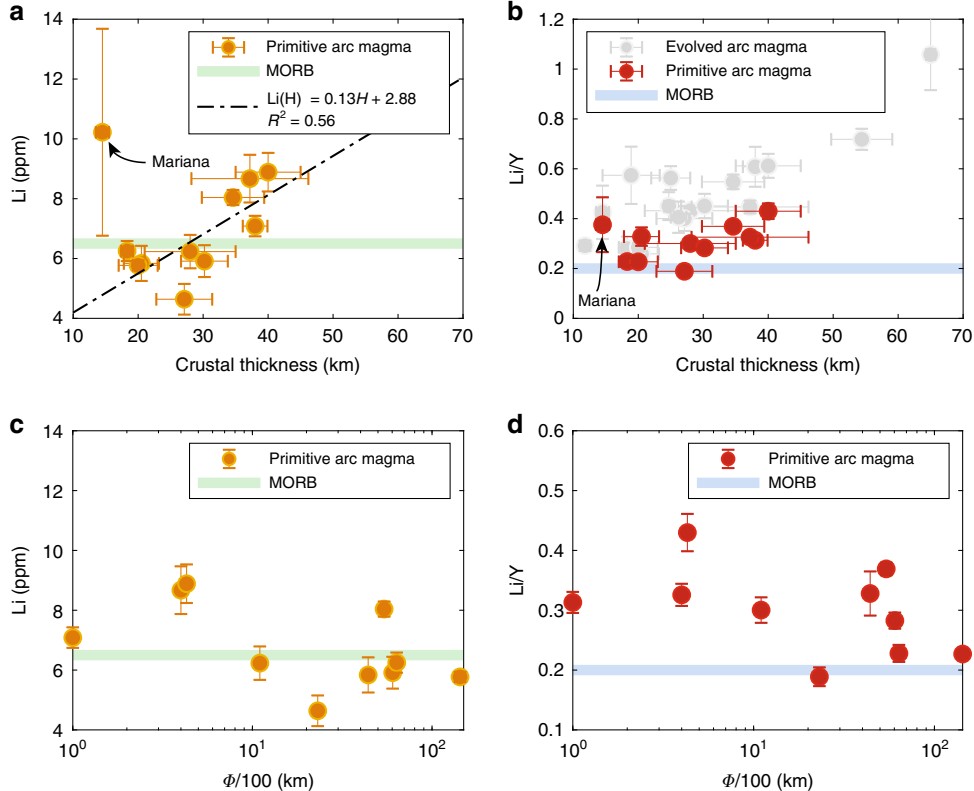

**Fig. 3 Li systematics of primitive arc magmas versus crustal thickness and slab thermal parameter. a**, **b** Li and Li/Y in primitive arc magmas vs. crustal thickness. Primitive arc magmas represent only those with 48–52 wt.% $SiO_2$ and 6–15 wt.% MgO. In (**b**), dull gray symbols represent averages for more evolved magmas (53–57 wt.% $SiO_2$). Error bars for Li concentration and Li/Y ratio are 2se; error bars for crustal thickness are one standard deviation ($1\sigma$) (Supplementary Dataset 2). As in Fig. 2, the pale green bar in (**a** and **c**) and blue bar in (**b** and **d**) show the average Li concentration of MORB from Gale et al.[22] and the Li/Y ratio of MORB from Ryan and Langmuir[9], respectively. The dashed line in (**a**) represents the linear regression of the data (with the Mariana arc excluded). **c**, **d** Li and Li/Y of primitive arc magmas vs. slab thermal parameter ($\Phi/100$). Slab thermal parameters are from Syracuse et al.[34].

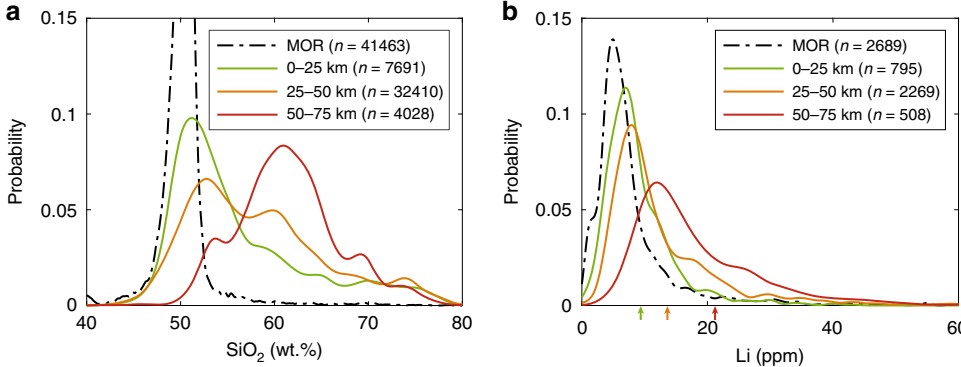

**Fig. 4 Effects of crustal thickness on magmatic differentiation and Li enrichment.** Probability distribution of $SiO_2$ (**a**) and Li concentrations (**b**) for arcs of different crustal thicknesses. Mid-ocean ridge (MOR) magmas shown as a reference. Colored arrows in (**b**) represent average Li concentrations for each crustal thickness group.

source rocks for further concentration of Li. We recognize the possibility of Li loss during magmatic degassing as well as during weathering after eruption may influence the whole-rock compositions[3,45] used here to investigate Li systematics. However, these effects would be most pronounced in subaerial environments and hence would affect evolved rocks more, making our enrichment factors minimum bounds.

We now explore the preferential occurrences of Li brine reservoirs. A first-order observation is that most Li brine deposits and clays occur in active continental arcs or magmatic orogens, not island arcs (Fig. 1a). Continental arcs and magmatic orogens,

both regions characterized by thick crust, may be critical for the formation of such deposits. Concomitant with the generation of more silicic magmas during crust thickening is the development of high orogenic plateaus[41,46], much like in the present-day central Andes, Tibetan plateau, and in the North American Cordillera during the Cretaceous. Intermontane, internally drained basins develop in these orogenic plateaus[47–49]. Combined with the high elevations, which generate rain shadows, many of these intermontane basins are net evaporative[4,50,51]. Ideal conditions for generating Li-rich brines may be met if these basins are surrounded by Li-rich source rocks, particularly easily

weatherable tuffs of andesitic or rhyolitic compositions. Leaching of these Li-enriched source rocks in the uplands by surface runoff or groundwater flow transports soluble Li into these confined basins, where it is further concentrated in the form of brines with progressive evaporation[1,4,51,52].

Basinal brine Li deposits thus represent the culmination of a series of compounding enrichment factors in the following order: (1) low degree melting in continental arc systems, (2) formation of evolved felsic magmas by fractional crystallization, (3) weathering of magmatic source rocks, and (4) transport and concentration of mobilized Li to internally drained, arid inter-montane basins. These properties are primarily found in magmatic orogens, which have undergone extensive crustal thickening (>60 km). Beyond the central Andes, the Tibetan plateau and the continental extensional belts, such as the Basin and Range in western USA, are ideal places to explore for basinal brines[1,4,51]. Within these thick arc systems, careful consideration of the distribution of Li-rich source rocks and intermontane basins should prove fruitful for exploration. For example, in continental arcs, such as the Andes, the combination of the most Li-enriched source rocks (rhyolites) and intermontane basins are typically found behind the main magmatic front[53]. Finally, given the association of Li with andesitic or rhyolitic source rocks, it may be worth exploring for Li in sedimentary basins that have accumulated significant amounts of ash, such as in the Cretaceous western interior seaway, a retro-arc basin formed during the peak of Cordilleran arc magmatism. Some oilfield brines are enriched in Li, and the possibility of a volcanic origin, in addition to marine evaporite origins, may be worth exploring[54,55].

In summary, we have shown that the contribution of Li from slab-related fluids or melts to arc magmas is small. Lithium enrichment in arc magmas instead is controlled by the conditions of melting in the mantle wedge and the extent of intracrustal differentiation. Arc magmas that traverse thick crust begin with higher initial Li contents due to lower extents of decompression melting and then evolve to higher Li contents through more extensive intracrustal differentiation. These differentiated magmas become ideal source rocks for the release of Li during weathering. This mobilized Li is eventually transported to land-locked intermontane basins, which are a unique characteristic of the tectonic environments responsible for making thick continental arcs. Exploration of basinal Li deposits beyond the Andes should focus on identifying these geologic and petrologic conditions in space and time.

## Methods

**Geochemical data**. To evaluate Li systematics of volcanic arcs on a global scale, we extracted geochemical data from the GEOROC database (http://georoc.mpch-mainz.gwdg.de/georoc/)[56]. Our compiled data cover nearly all active magmatic arcs on Earth, from island arcs to continental arcs. The sample locations are plotted in Supplementary Fig. 4. We filtered out plutonic, sedimentary, and metamorphic rocks, retaining only volcanic whole-rock data to ensure that the samples are representative of recent magmatism. Samples with major element oxide totals outside the range 98–101.5 wt.% were excluded in order to minimize alteration effects. The final arc database is available in Supplementary Dataset 3. We are particularly interested in Li concentrations and Li/Y ratios in arc magmas. When plotting Li and element ratios versus $SiO_2$ content, we first binned the samples by $SiO_2$ content. Within each $SiO_2$ bin, we removed samples that fell in the upper and lower 10% of the distribution to minimize outlier effects. We then calculated averages and 2 standard errors (2 standard deviation divided by the square root of the number of samples) for each bin as long as the number of samples was greater than 5. Element ratios, such as Li/Y, were estimated by ratioing the average values of Li and Y in a bin and by directly averaging sample Li/Y ratios; both approaches yield consistent results (Supplementary Dataset 4 and Supplementary Fig. 5) so only the latter are presented. When investigating magma sources of arc segments, we define primitive magmas as those with 48–52 wt.% $SiO_2$ and 6–15 wt.% MgO to minimize differentiation effects.

**Geophysical parameters**. We also investigated Li systematics of young volcanic rocks as functions of crustal thickness. Generalized crustal thicknesses for specific

arc segments were obtained from seismically constrained Moho depths (Supplementary Dataset 2). However, seismic studies do not cover all active arcs, and there can be significant within-arc crustal thickness variations. To improve our data coverage, we estimated crustal thickness from present-day elevation because on long enough length scales (>10 km), the crust in arcs is isostatically compensated due to the hot and weak deep crust beneath active arcs, as documented by the correlation of elevations of mountains versus Moho depths[46]. Using an empirical correlation between elevation and crustal thickness[41,46], we can assign a crustal thickness to the location from which individual rocks were sampled. We obtained the elevations of geo-located samples from the ESRI ArcGIS software package. Following the approach of Farner and Lee[41], elevations for given locations were extracted from the National Oceanographic and Atmospheric Administration ETOPO2 digital elevation model (https://www.ngdc.noaa.gov/mgg/global/etopo2)[57], which we modified in ArcGIS to generate 20 km × 20 km grids to eliminate non-isostatic topography.

**Calculation and simulation of Li enrichment during crystallization**. We modeled how Li becomes enriched by batch crystallization. We adopted a bulk partition coefficient (D) of ~0.2 based on prior natural and experimental studies[13–17]. Lithium contents in residual melts ($C_{Li\ 1}$) are calculated as

$$C_{Li1} = \frac{C_{Lio}}{D(1-F)+F}, \tag{1}$$

where $C_{Li\ o}$ is the initial Li concentration in parental magmas, which we assume to be the same as the average Li contents of primitive magmas in arcs of different crustal thickness. $F$ represents relative residual melt fraction.

One approach for assessing $F$ is to obtain the correlation between $SiO_2$ content of residual magma and corresponding $F$. Using Rhyolite-MELTS[58], we simulated crystal fractionation to generate this correlation, which we then used to convert $SiO_2$ content into $F$. The starting composition is set as the average primitive magma composition of a corresponding arc. Specific starting compositions and simulation conditions are provided in Supplementary Dataset 5 and Supplementary Dataset 6.

Another way to estimate $F$ is based on incompatible element concentrations of given samples. For a perfectly incompatible element, $F$ is the inverse of enrichment of any derivative magma ($C_l$) relative to its initial parent composition ($C_o$), that is,

$$F = \frac{C_o}{C_l}. \tag{2}$$

We used potassium here because of its nearly perfect incompatibility[18–20]. Initial K contents ($C_o$) of each group are adopted from the average K concentration of primitive arc basalts.

## Data availability

The compilation of lithium resources and reserves is provided in Supplementary Dataset 1. Li and Li/Y statistics for arc segments are provided in Supplementary Dataset 2. The compiled global arc whole rock data extracted from GEOROC[56] are provided in Supplementary Dataset 3. The comparison between different methods of calculating elemental ratios are provided in Supplementary Dataset 4. The starting composition input for Rhyolite-MELTS[58] simulation is provided in Supplementary Dataset 5 and simulation conditions are in Supplementary Dataset 6.

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

## Acknowledgements

The authors thank Ralf Halama, Paul Tomascak for constructive suggestions. We also thank Patrick Phelps, Hehe Jiang, and Sydney Allen for discussions. Shengpan Wang from Wuhan University and Michael Farner are thanked for help with ArcGIS. This work was supported by the US National Science Foundation (NSF EAR—1850832) to Cin-Ty A. Lee and Ming Tang and a China Scholarship Council award to Chen Chen (File No. 201704910773).

## Author contributions

C.C. compiled the data and carried out the simulations; C.-T.A.L. and C.C. initiated the project; C.C., C.-T.A.L. and M.T. interpreted the data and wrote the manuscript; K.B.

provided the ore data compilation; K.B. and W.S. provided comments and suggestions that improved the manuscript.

## Competing interests

The authors declare no competing interests.
