## [Peer Review File · Nature Communications]

Reviewers' comments:

Reviewer #1 (Remarks to the Author):

This manuscript describes the preferential enrichment of lithium in continental arcs relative to oceanic arcs based on a compilation of global data, and links the results to lithium exploration. The manuscript nicely demonstrates the relevance of magmatic geochemistry/petrology for general aspects of the occurrence of ores and as such exemplifies how geochemistry plays a role for issues of societal relevance such as the availability of lithium for the battery industry. Thematically, the manuscript is well suited for Nature Communications. The data compilation is sound, but in some instances some more detail and/or a more quantitative treatment/modelling would benefit the interpretation. I like the bold approach to use a carefully selected and filtered global dataset for this study and realize that this does require some simplification in the main text, but some expansion on certain aspects, which could be presented in a supplementary discussion, would be useful.

Detailed comments:

Abstract: The abstract emphasizes the occurrence of lithium from basinal brines and the links of magmatism to hydrology, but does not mention Li-rich pegmatites, which are important later in the text. This should be more consistent: Either the pegmatites should be mentioned in the abstract (as it is well known that they are an important source of Li), or, alternatively, the paper only focusses on Li from brines. The pegmatites that are mentioned in some detail (California, Black Hills) rather complicate the overall picture and do only partially support the general statements made, so if pegmatites are included it would be useful to provide examples that support the general ideas.

Line 38-41: A few references should be given in relation to the occurrence of Li deposits in the various settings (pegmatites, evaporative basins, but also clays and oilfields).

Line 55-58: Whereas Li in the Andes is in a continental arc setting, the Li brines from Tibet/China are in a collisional orogen. Hence, collisional settings also appear important for thickening the crust and enriching Li in the magmatic rocks generated, so this should be more explicitly considered.

Line 76-78: Li-bearing pegmatites in continental arcs: The statement that Li-bearing pegmatites occur in more juvenile stages of continental arcs needs to be expanded upon: Why is that the case? Does that not, at least to some degree, contradict the idea that thick, mature crust is necessary to facilitate Li-enrichment? Many of the pegmatite locations on the map occur in Precambrian basement, and it is not immediately clear what kind of tectonic environment this basement represents: Which of the pegmatites occur in basement that is representing continental arcs, and how is the link to juvenile arcs in the Precambrian basement justified?

Line 84: Incompatible element: Support this statement with referencing work on Li partitioning, e.g.

Brenan et al. 1998, *Geochim. Cosmochim. Acta* 62
Ottolini et al., 2009, *Contrib. Mineral. Petrol.* 157

99-101: Differentiation in thick arcs: This statement seems to imply that differentiation in thick arcs occurs at high pressures. However, many arc magmas undergo extensive differentiation in relatively shallow magmatic reservoirs, so the implication is not well supported (or the argument is not clearly formulated?). To show effects of high-P differentiation, one would presumably select rocks that show such a geochemical signature. Note also that there is a jump to higher Li/K values in thin arcs for 68-75 wt% SiO₂ – please provide an explanation for this feature.

109-110: Ok, but change the wording: Avoid „primitive Li“, rather use „Li in primitive magmatic rocks“. What is „primitive Li/Y“?

111-113: Sentence is not clear, please reformulate. Are amphibole and garnet melted (i.e. elements will be incorporated into melt) or are the elements sequestered into these minerals (i.e. removed from the melt)?

111-120: Details of the REE should also be explained in more detail: What does a high Dy/Yb indicate? Since garnet and amphibole have distinct REE patterns, the effects of each of those in the source should be more clearly explained. Amphibole, in general, also incorporates MREE, whereas garnet has a clear preference for HREE. Also, provide references for Li partitioning in garnet, e.g.:

van Westrenen et al., 1999, *American Mineralogist* 84: 838-847.

Note also that charge-coupling between Li⁺ and REE³⁺ have been observed in garnet, causing Li profiles to look similar to Y and REE profiles:

Carlson, *Int. Geol. Rev.* 59: 526-540.

Overall, this section would benefit from a more detailed discussion and clarification of arguments.

157: enriched relative to? Add to what this is compared to.

181-188: High-P fractionation and role of garnet: A model of fractional crystallisation with the relevant mineral phases, including garnet, would be useful to support your argument here. Quantification of the „incompatibility“ of Li in garnet based on experimental partitioning data, and modelling the results could show the reader that garnet fractionation can indeed increase Li relative to other elements. I agree with the statements made, but they are rather qualitative and in this manuscript, where this process is fundamental to the story of Li enrichment, a quantitative treatment is desirable.

190: typo „intracrustal“

207-208: Orogenic plateaus: The Tibet plateau, which shows Li reserves on the map, should also be considered here. Some features are similar to continental arcs (thick crust, high elevation, rain shadow), but it is formed in a collisional setting rather than arc setting.

220-221: „may be ideal places“ – why so vague? There seem to be Li brine deposits in these areas on your map, so they are already explored for Li. Be more detailed and check and cite references that would help to support your argument, e.g.:

Gao et al., 2012: *Aquatic Geochemistry* 18: 343–356

225: Western interior seaway: Explain (to an international audience) what and where that feature is.

227-249:

Overall, this section does not support your main line of argumentation. In contrast, examples are mentioned where more Li-rich pegmatites are found in association with more mafic magmas (California), or where Li-rich pegmatites are not related to igneous host rocks (New Mexico, Black Hills). Are there any Li-rich pegmatites associated with thick continental arc crust? What is the relative abundance of those compared to other settings? How were those in Precambrian terranes classified? Either the pegmatites require a more detailed consideration, or consider whether shifting the focus away from pegmatites might be useful in formulating a clearer argumentation.

Figures:

1A: This is a nice map, but in the printed version, geographical details are hard to see, in particular outlines of continents (e.g. the dots in Cornwall are clear, but one can only guess the outline of Britain). Consider modifying the contrast/overlay colours.

1B: Coordinates are very small and should be increased in font size. Also, why are no Li resources shown in China and the western US, even though reserves are clearly present? As by definition the resources include the total Li available, the resource dots should be larger than those for the reserves.

References:

24 – Check journal. Use different reference if journal is unknown.

R. Halama (Keele University), 2/2/2020

Reviewer #2 (Remarks to the Author):

Review of Nature Communications manuscript by H-B Choi et al., "Lithium systematics in global arc magmas 1 and implications for exploration"

Review by Paul Tomascak, SUNY Oswego, 2/9/20

The manuscript presents two main arguments. The first is that Li enrichment in subduction settings has been exaggerated by prior authors. The second is that Li enrichment should be most prominent in continental arcs, or where crust is thick, and that this enrichment is the ultimate feedstock for the current major crustal Li deposits in brines and similar evaporitic basinal settings.

Argument one is reasoned rather well by examining only the low-Si ("primitive") samples from arcs. Prior loud pronouncements (notably Plank, 2014) discuss Li in arcs as having a substantial (dominant?) source in slab-derived fluids. I will point out that our Li isotope work has largely argued that arc rocks do not show a significant slab-Li signature (e.g., Tomascak et al., 2002); I admit surprised that this is not mentioned in the manuscript. Further, there is a rather unimpeachable example of slab-Li being present in at least one arc setting (Martinique, Tang et al. 2014). Thus I think this may be a little more complicated than the authors present, though in general I think the argument has broad justification.

The second argument carries reasonable support, though I think the authors go a bit astray in their presentation of assertions about granitic pegmatites. The oddity of highly Li-enriched granitic pegmatites in thin-crust arc environments is not presented fairly, I think, which is quite odd. The San Diego County pegmatites are real outliers in global terms, a point the manuscript makes in only a quite backwards way and with no reference to the two places in North America where Li has been mined from hard rock in the last several decades (NC and Manitoba).

All in all I think the first argument is somewhat important, though the lack of folding in Li isotope support is peculiar. The second argument is a bit of a so-what, I think. Yes, we will need to keep searching for new Li deposits but I don't see the conclusions reached here as in any way revelatory. It is a good, brief summing up of what I think is already pretty obvious.

I have some comments related to individual parts of the text below.

line 5: The deposits are brines; to say they are found in brines seems odd wording.

line 10: As always, avoid beginning sentences with abbreviations.

line 52: "...are similar to those of... higher than those of MORB..."

line 75: You previously just used "andesitic source rocks" which I think is more appropriate than a specific rock type like this.

line 85: "to that of MORB (6.5 ppm..."

line 87: "extreme alteration"? Really?

line 94, 95: Avoid possessives: "Pressure effects on bulk partitioning of Li."

line 100: "...with differentiation have been observed..."

line 106: "thought to be fluid mobile"? How about citing a paper that demonstrates it is.

line 109: "as widely thought"... Well, I see what you are trying to do, but does one paper make something "widely thought"?

line 111: "Such a slight correlation can be interpreted to result from melting residual amphibole and garnet, which sequester Y, consistent with..."

line 117: Avoid using possessives.

line 119: The Dy/Yb argument seems to need to go in the prior paragraph.

line 171: "...has been interpreted as the effect of..."

line 185: Sun et al. (Chem Geol 2016) report garnet with high Li concentrations. Coupled with generally low Li contents in anorthosites, I think the garnet/plagioclase argument needs to be better supported.

line 189: Pick on: degree or extent. You use them interchangeably in the prior text.

line 190: Intracrustal is misspelled.

line 221: "may be ideal places"? There is literature to support this for both locations.

line 222: It has been all andesites up to this point, so why suddenly introduce rhyolite? Be consistent.

line 228: This definition of pegmatite is a bit silly/overly simple.

line 229: "Li-rich ores are found only in highly silicic (granitic) pegmatites..." Misleading (as only a very small proportion of granitic pegmatites are enriched in Li to ore grade) and inaccurate (non-pegmatite deposits like greisens are also Li-mineralized).

line 231: Considering a basaltic "parent" for Li-rich pegmatites is perhaps worthwhile from a purely conceptual standpoint, but there is not a Li-mineralized pegmatite out there that descends on a straight line from a basaltic magma.

line 233: "should also lead to enrichment in volatiles" with one reference. There are plenty of perhaps better examples; more appropriate to give this a "will lead to similar enrichment in volatiles."

line 238: Bringing up this example really neglects the forest for the trees in this argument: the vast majority of Li-mineralized pegmatites are hosted by thick crust.

line 246-249: These last two sentences expose the fallacy of this argument. Harding and the Black Hills are the norm for Li-enriched granite deposits. Treating them like the odd balls is completely backwards.

Reviewer #3 (Remarks to the Author):

This paper uses existing data to discuss the systematics of Li in arc magmas. It does a good job in summarizing what is already known about Li enrichment: it is a phenomenon only associated with magmas which have traversed continental crust. The paper is more of a summary paper, as no new data is presented. The conclusions reached are valid, though the exploration and deposit formation components in the discussion are seemingly disjointed from the geochemical aspects of the paper, which should be the main focus.

The conclusions - especially with regard to the exploration and deposit formation side of things - are unoriginal and have been discussed extensively by previous authors (Godfrey, Munk, etc.) These works - aside from one Munk paper - are not referenced. The authors present no data, evidence, or figures to support their deposit formation models. There are papers that have done this extensively, and I think the focus of this paper should remain on the behavior of Li in the deep crust.

The focus on deep crustal processes leading to small 10s of ppm variations in Li concentration is interesting but not a major factor in Li deposit formation, therefore this paper should not focus on Li deposits but discuss deep crustal Li variation in greater detail. The authors discuss later on in the paper that upper crustal differentiation plays a major role in the formation of Li deposits yet the paper does not discuss rhyolite data. There is also no mention of specific lithium deposits in the whole paper or the magmas/volcanic rocks associated with them, which I think is a critical

piece to discussing Li deposits - especially exploration for them. E.g. Atacama, Hombre Muerto, Thacker Pass, Greenbushes. It is not sufficient to compile geochemical data and not relate them to specific lithium deposits (or even mention a Li deposit) in a paper about Li deposits.

The authors also need to do some more research on actual pegmatite deposits and how they form -there is a lot of data out there on the mechanisms involved in how they form. Even looking at the geological maps of the biggest hard rock resources in the world (Greenbushes, Pilgangoora, Tanco, etc.) one could see that the association with metapelites is ubiquitous. Extensive work has been done in this regard - there is a large volume of pegmatite literature out there (see London, among others).

I think there is interesting information in this paper but the authors need to rethink the purpose of the manuscript. I think it has a great contribution to understanding Li mobility in the deep crust but the extrapolation to Li resources is unresearched, unreferenced, and therefore unfounded.

Line by line comments:

Line 15. Nearly all lithium deposits are sourced from rhyolites and to a lesser extent, dacites, not andesites. In brine systems, 100% of producing brine deposits have rhyolitic tuff in the watershed. Salars without rhyolitic tuffs in the watershed do not have lithium deposits. In clay/sedimentary lithium deposits in the Basin and Range, lithium is sourced from rhyolite tuffs as well.

Line 17. Most brine deposits occur in back arc extensional basins and into intracontinental settings, not in the continental arc itself. I don't see any discussion of any actual lithium deposits or associated magmas in this paper. I think this needs to be changed.

Line 19. Change to, "the development of Li-enriched brines is thus..."

Line 36. Should be, "Growth in DEMAND of hybrid...."

Line 40. Li clay/sedimentary deposits are actually quite large. The Thacker Pass Project in NV is the 4th largest Li resource in the world. Oilfield brines and geothermal brines are inconsequential and the technology is not there to extract them yet.

FIGURE 1A. This graph mainly just shows degree of erosion. Precambrian basement = old and therefore the pegmatites are exposed.

FIGURE 1B and 1C. These plots are extremely hard to read.

FIGURE 1D. Rhyolite tuff, not andesite tuff.

Line 76. Partial melting of pelitic metamorphic rocks is a major source for pegmatites. Look at geological maps of the main deposits - the spodumene, Li-bearing pegmatites are nearly ubiquitously intruding these rocks.

FIGURE 2. Trends are there, but do not play a major role in what magma actually leads to a lithium deposit. A change in approximately 20 ppm Li in these magmas is inconsequential relative to what happens in magma chambers within the upper crust. The lithium concentration can vary 100s of ppm in a single rhyolitic magma chamber due to differentiation processes. That plays way more of a role than these few - 10s of ppm changes in the deep crust. It has long been known that Li is higher on continental crust.

Line 114-120. This is interesting, but really just a commentary on the compatibility of Y. Li is increasing with SiO₂ content regardless of the Y.

FIGURE 3. Is this plot necessary? This really is just showing that the degree of upper crustal differentiation is the main control on Li concentration, not slab differences.

Line 163. I think it is widely accepted that Li is coming from the crust, not the slab. Most Li deposits in the world occur from the back arc to intracontinental settings.

Line 164. Once again, this is dealing in the 10s of ppms. It is inconsequential compared to processes in the upper crust which have a much greater control on Li enrichment. In a given tectonic setting, say in a back arc, if you have an andesite, there will be less than 50 ppm Li in the magma. If that same magma evolves, there could be 10x the amount of Li. Volcanic rocks that are sources for the main Li brine deposits in the Li Triangle have Li contents that vary by a factor of 5 within individual eruptions, consistent with fractional crystallization. The extent to which these magmas are within the watersheds of the salars is the controlling factor in the formation of Li deposits. The 10-20 ppm variations in arc magmas might play a tiny role very early on but it is inconsequential compared to this. Look at any non-producing salar in the Li triangle. None of them have dacite - rhyolite tuffs, most of them have andesites in their watersheds.

Line 184. Many Li deposits and Li-rich magmas are not associated with arc magmatism, though.

Line 205. This was the thesis of Benson et al. (2017)

Line 209. This is extensively studied and known. See work by Godfrey and Munk, among others. (Godfrey et al 2013), Munk et al (2016), Munk et al (2018). This work needs to be credited.

Line 215. These processes, documented in detail by other workers, are the main mechanisms leading to Li brine deposits – and really are disconnected from the data presented earlier.

Line 240. The data is out there.

Line 253. This is not the main control.

Revision notes

We greatly appreciate the detailed and constructive comments from three reviewers. We agree with the reviewers and revised our manuscript accordingly. Primarily,

- We have removed discussion about pegmatites and focused the manuscript primarily on understanding the nature by which Li is enriched in arc magmas as suggested by reviewers;
- We now show model calculations of Li enrichment during crystal fractionation, explaining how thickened arcs undergo greater Li enrichments than thin arcs (a new figure panel in Fig. 2 has been added to show model results);
- We have provided more details on how we estimated residual melt fraction in the Methods section;
- As suggested by reviewers, we have added a number of key references;
- We have also fixed any typos and modified sentences to be clearer.

Our point-by-point responses are below. Specific changes in the manuscript itself are highlighted in red.

Reviewer #1 (Remarks to the Author):

This manuscript describes the preferential enrichment of lithium in continental arcs relative to oceanic arcs based on a compilation of global data, and links the results to lithium exploration. The manuscript nicely demonstrates the relevance of magmatic geochemistry/petrology for general aspects of the occurrence of ores and as such exemplifies how geochemistry plays a role for issues of societal relevance such as the availability of lithium for the battery industry. Thematically, the manuscript is well suited for Nature Communications. The data compilation is sound, but in some instances some more detail and/or a more quantitative treatment/modelling would benefit the interpretation. I like the bold approach to use a carefully selected and filtered global dataset for this study and realize that this does require some simplification in the main text, but some expansion on certain aspects, which could be presented in a supplementary discussion, would be useful.

Thank you for your constructive comments. We originally did not think it was necessary to model as we felt that the data spoke for themselves. However, we have followed this suggestion and now present the results of fractional crystallization modeling in Figure 2. Details are now provided in supplementary text.

Detailed comments:

Abstract: The abstract emphasizes the occurrence of lithium from basinal brines and the links of magmatism to hydrology, but does not mention Li-rich pegmatites, which are important later in the text. This should be more consistent: Either the pegmatites should be mentioned in the abstract (as it is well known that they are an important source of Li), or, alternatively, the paper only focusses on Li from brines. The pegmatites that are mentioned in some detail (California, Black Hills) rather complicate the overall picture and do only partially support the general statements made, so if pegmatites are included it would be useful to provide examples that support the general ideas.

To keep our manuscript focused, we have removed the pegmatite section. Abstract and remaining text have been adapted accordingly.

Line 38-41 (now in line 35-39): A few references should be given in relation to the occurrence of Li deposits in the various settings (pegmatites, evaporative basins, but also clays and oilfields).

We have now updated the references.

Line 55-58 (now in line 39-41): Whereas Li in the Andes is in a continental arc setting, the Li brines from Tibet/China are in a collisional orogen. Hence, collisional settings also appear important for thickening the crust and enriching Li in the magmatic rocks generated, so this should be more explicitly considered.

We have now rephrased the relevant sentences.

Line 76-78 (now removed): Li-bearing pegmatites in continental arcs: The statement that Li-bearing pegmatites occur in more juvenile stages of continental arcs needs to be expanded upon: Why is that the case? Does that not, at least to some degree, contradict the idea that thick, mature crust is necessary to facilitate Li-enrichment? Many of the pegmatite locations on the map occur in Precambrian basement, and it is not immediately clear what kind of tectonic

environment this basement represents: Which of the pegmatites occur in basement that is representing continental arcs, and how is the link to juvenile arcs in the Precambrian basement justified?

Because we have now eliminated the discussion on pegmatites, this comment no longer applies.

Line 84 (now in line 78): Incompatible element: Support this statement with referencing work on Li partitioning, e.g. Brenan et al. 1998, *Geochim. Cosmochim. Acta* 62
Ottolini et al., 2009, *Contrib. Mineral. Petrol.* 157

We have now updated these references.

99-101 (now in line 78-81): Differentiation in thick arcs: This statement seems to imply that differentiation in thick arcs occurs at high pressures. However, many arc magmas undergo extensive differentiation in relatively shallow magmatic reservoirs, so the implication is not well supported (or the argument is not clearly formulated?). To show effects of high-P differentiation, one would presumably select rocks that show such a geochemical signature. Note also that there is a jump to higher Li/K values in thin arcs for 68-75 wt% SiO₂ – please provide an explanation for this feature.

We have modified this section as we see how the original version may have confused readers. The dominant effect of thick crust is to drive more extensive differentiation, which is integrated through the entire crustal column, from the deep crust to the shallow crust as the reviewer notes. The direct effect of deep crust or high P fractionation is only a secondary effect, which we state more clearly now. This can be seen in Li/Y ratio differences, but only slight differences in Li/K.

109-110 (now in line 101-103): Ok, but change the wording: Avoid „primitive Li“, rather use „Li in primitive magmatic rocks“. What is „primitive Li/Y“?

We have now rephrased accordingly.

111-113 (now in line 104-106): Sentence is not clear, please reformulate. Are amphibole and garnet melted (i.e. elements will be incorporated into melt) or are the elements sequestered into these minerals (i.e. removed from the melt)?

Rephrased accordingly.

111-120 (now in line 104-115): Details of the REE should also be explained in more detail: What does a high Dy/Yb indicate? Since garnet and amphibole have distinct REE patterns, the effects of each of those in the source should be more clearly explained. Amphibole, in general, also incorporates MREE, whereas garnet has a clear preference for HREE. Also, provide references for Li partitioning in garnet, e.g.:
van Westrenen et al., 1999, *American Mineralogist* 84: 838-847.
Note also that charge-coupling between Li⁺ and REE³⁺ have been observed in garnet, causing Li profiles to look similar to Y and REE profiles:
Carlson, *Int. Geol. Rev.* 59: 526-540.
Overall, this section would benefit from a more detailed discussion and clarification of arguments.

We have reduced the discussion on Dy/Yb and placed into supplementary text. We have explained in more detail the different effects of gt and amphibole on Dy/Yb as well as on Li/Y.

157 (now in line 161-162): enriched relative to? Add to what this is compared to.

Corrected.

181-188 (now in line 92-94): High-P fractionation and role of garnet: A model of fractional crystallization with the relevant mineral phases, including garnet, would be useful to support your argument here. Quantification of the „incompatibility“ of Li in garnet based on experimental partitioning data, and modelling the results could show the reader that garnet fractionation can indeed increase Li relative to other elements. I agree with the statements made, but they are rather qualitative and in this manuscript, where this process is fundamental to the story of Li enrichment, a quantitative treatment is desirable.

Crystal fractionation model has been performed as suggested. This is now in the paper.

190 (now in line 192): typo „intracrustal“

Fixed.

207-208 (now in line 208-209): Orogenic plateaus: The Tibet plateau, which shows Li reserves on the map, should also be considered here. Some features are similar to continental arcs (thick crust, high elevation, rain shadow), but it is formed in a collisional setting rather than arc setting.

Fixed.

220-221 (now in line 222-224): „may be ideal places“ – why so vague? There seem to be Li brine deposits in these areas on your map, so they are already explored for Li. Be more detailed and check and cite references that would help to support your argument, e.g.:
Gao et al., 2012: Aquatic Geochemistry 18: 343–356

Fixed. We have now added relevant references.

225 (now in line 227): Western interior seaway: Explain (to an international audience) what and where that feature is.

Clarified.

227-249 (now removed):

Overall, this section does not support your main line of argumentation. In contrast, examples are mentioned where more Li-rich pegmatites are found in association with more mafic magmas (California), or where Li-rich pegmatites are not related to igneous host rocks (New Mexico, Black Hills). Are there any Li-rich pegmatites associated with thick continental arc crust? What is the relative abundance of those compared to other settings? How were those in Precambrian terranes classified? Either the pegmatites require a more detailed consideration, or consider whether shifting the focus away from pegmatites might be useful in formulating a clearer argumentation.

We have removed pegmatite discussions, so this comment is not applicable anymore.

Figures:

1A: This is a nice map, but in the printed version, geographical details are hard to see, in particular outlines of continents (e.g. the dots in Cornwall are clear, but one can only guess the outline of Britain). Consider modifying the contrast/overlay colours.

We have modified the figure to make it more readable.

1B: Coordinates are very small and should be increased in font size. Also, why are no Li resources shown in China and the western US, even though reserves are clearly present? As by definition the resources include the total Li available, the resource dots should be larger than those for the reserves.

Fixed.

References:

24 – Check journal. Use different reference if journal is unknown.

Fixed.

R. Halama (Keele University), 2/2/2020

Reviewer #2 (Remarks to the Author):

Review of Nature Communications manuscript by Chen et al., "Lithium systematics in global arc magmas and implications for exploration"

Review by Paul Tomascak, SUNY Oswego, 2/9/20

The manuscript presents two main arguments. The first is that Li enrichment in subduction settings has been exaggerated by prior authors. The second is that Li enrichment should be most prominent in continental arcs, or where crust is thick, and that this enrichment is the ultimate feedstock for the current major crustal Li deposits in brines and similar evaporitic basinal settings.

Argument one is reasoned rather well by examining only the low-Si ("primitive") samples from arcs. Prior loud pronouncements (notably Plank, 2014) discuss Li in arcs as having a substantial (dominant?) source in slab-derived fluids. I will point out that our Li isotope work has largely argued that arc rocks do not show a significant slab-Li signature (e.g., Tomascak et al., 2002); I admit surprised that this is not mentioned in the manuscript. Further, there is a rather unimpeachable example of slab-Li being present in at least one arc setting (Martinique, Tang et al. 2014). Thus I think this may be a little more complicated than the authors present, though in general I think the argument has broad justification.

We are embarrassed that we forgot to cite the above-mentioned papers. That was an oversight. These references are now included, along with more discussion. We point out now that in some cases, Li isotopes does show a slab signature, but the majority do not as you say. The Li/Y systematics are probably more diagnostic given the many different processes that influence Li isotopes. We also address briefly the reason why Plank came up with high Li/Y.

The second argument carries reasonable support, though I think the authors go a bit astray in their presentation of assertions about granitic pegmatites. The oddity of highly Li-enriched granitic pegmatites in thin-crust arc environments is not presented fairly, I think, which is quite odd. The San Diego County pegmatites are real outliers in global terms, a point the manuscript makes in only a quite backwards way and with no reference to the two places in North America where Li has been mined from hard rock in the last several decades (NC and Manitoba).

We removed pegmatite discussion, so this issue is not applicable anymore.

All in all I think the first argument is somewhat important, though the lack of folding in Li isotope support is peculiar. The second argument is a bit of a so-what, I think. Yes, we will need to keep searching for new Li deposits but I don't see the conclusions reached here as in any way revelatory. It is a good, brief summing up of what I think is already pretty obvious.

I have some comments related to individual parts of the text below.

line 5 (in line 6): The deposits are brines; to say they are found in brines seems odd wording.

We have now rewritten this sentence.

line 10 (in the same line): As always, avoid beginning sentences with abbreviations.

Fixed.

line 52 (now rewritten): "...are similar to those of... higher than those of MORB..."

Fixed.

line 75 (now in line 70): You previously just used "andesitic source rocks" which I think is more appropriate than a specific rock type like this.

Fixed.

line 85 (now in line 88): "to that of MORB (6.5 ppm..."

Fixed.

line 87 (now in line 84-86): "extreme alteration"? Really?

The Mariana whole rock data are scattered particularly at the primitive end. We went back to the original papers that contain Li data to check their petrography (Dietrich et al., 1978; Sharaskin, 1982). They were reported to be highly altered basaltic rocks dredged from the West Mariana basin. The secondary minerals are smectite, montmorillonite, chlorite, serpentine minerals etc., which are commonly enriched in water and fluid-mobile elements. Li concentrations vary considerably for the low SiO₂ Marianas endmember. We also compare the Marianas itself with the global compilation of data for thin arcs (see figure below). The fact that Marianas also deviates from the overall trend at low SiO₂ forced us to conclude that we could not use the primitive Li contents of the Marianas.

line 94, 95 (now rewritten): Avoid possessives: “Pressure effects on bulk partitioning of Li.”

Fixed.

line 100 (now rewritten): “...with differentiation have been observed...”

Fixed.

line 106 (in line 99): “thought to be fluid mobile”? How about citing a paper that demonstrates it is.

We have now added a reference to support this point.

line 109 (in line 102): “as widely thought”... Well, I see what you are trying to do, but does one paper make something “widely thought”?

We have now added additional references. The elevated Li/Y in arc magma is ubiquitous as reported by prior studies (Brenan et al., 1998; Elliott, 2003; Elliott et al., 2004; Plank, 2014; Tomascak et al., 2002). This is due to Li/Y fractionation during magmatic differentiation as shown in Figure 2d rather than slab fluid contribution.

line 111 (in line 104): “Such a slight correlation can be interpreted to result from melting residual amphibole and garnet, which sequester Y, consistent with...”

Fixed.

line 117 (now in line 112): Avoid using possessives.

Fixed.

line 119 (now in line 113): The Dy/Yb argument seems to need to go in the prior paragraph.

We have done corresponding adjustments and added necessary discussion in Supplementary information.

line 171 (now in line 179): "...has been interpreted as the effect of..."

Fixed.

line 185 (now rewritten): Sun et al. (Chem Geol 2016) report garnet with high Li concentrations. Coupled with generally low Li contents in anorthosites, I think the garnet/plagioclase argument needs to be better supported.

Thanks for suggestions. We have now conducted new modeling to document how initial Li contents and Li incompatibility impose on Li enrichment at different thickness arcs. However, we note in the revision that the garnet effect is small after all.

line 189 (now in line 191): Pick on: degree or extent. You use them interchangeably in the prior text.

Fixed.

line 190 (now in line 192): Intracrustal is misspelled.

Fixed.

line 221 (now in line 223): "may be ideal places"? There is literature to support this for both locations.

We have added relevant papers.

line 222 (now in line 224): It has been all andesites up to this point, so why suddenly introduce rhyolite? Be consistent.

Clarified now. Andesites and rhyolites are now both discussed throughout the text.

line 228 (now removed): This definition of pegmatite is a bit silly/overly simple.

We dropped pegmatite discussion, so no longer applicable.

line 229 (now removed): "Li-rich ores are found only in highly silicic (granitic) pegmatites..." Misleading (as only a very small proportion of granitic pegmatites are enriched in Li to ore grade) and inaccurate (non-pegmatite deposits like greisens are also Li-mineralized).

We dropped pegmatite discussion, so no longer applicable.

line 231 (now removed): Considering a basaltic "parent" for Li-rich pegmatites is perhaps worthwhile from a purely conceptual standpoint, but there is not a Li-mineralized pegmatite out there that descends on a straight line from a basaltic magma.

We dropped pegmatite discussion, so no longer applicable.

line 233 (now removed): "should also lead to enrichment in volatiles" with one reference. There are plenty of perhaps better examples; more appropriate to give this a "will lead to similar enrichment in volatiles."

We dropped pegmatite discussion, so no longer applicable.

line 238 (now removed): Bringing up this example really neglects the forest for the trees in this argument: the vast majority of Li-mineralized pegmatites are hosted by thick crust.

We dropped pegmatite discussion, so no longer applicable.

line 246-249 (now removed): These last two sentences expose the fallacy of this argument. Harding and the Black Hills are the norm for Li-enriched granite deposits. Treating them like the odd balls is completely backwards.

We dropped pegmatite discussion, so no longer applicable.

Reviewer #3 (Remarks to the Author):

This paper uses existing data to discuss the systematics of Li in arc magmas. It does a good job in summarizing what is already known about Li enrichment: it is a phenomenon only associated with magmas which have traversed continental crust. The paper is more of a summary paper, as no new data is presented. The conclusions reached are valid, though the exploration and deposit formation components in the discussion are seemingly disjointed from the geochemical aspects of the paper, which should be the main focus.

We agree with the reviewer here. We have focused now primarily on the geochemical aspects rather than the economic geology. This should make the paper easier to digest and less disjointed.

The conclusions - especially with regard to the exploration and deposit formation side of things - are unoriginal and have been discussed extensively by previous authors (Godfrey, Munk, etc.) These works - aside from one Munk paper - are not referenced. The authors present no data, evidence, or figures to support their deposit formation models. There are papers that have done this extensively, and I think the focus of this paper should remain on the behavior of Li in the deep crust.

We have now cited more relevant papers. We have modified the paper so that the focus is on the behavior of Li in the slab, mantle and crust, and removed much of the earlier discussion on how these are distributed and transported in the surface environment.

The focus on deep crustal processes leading to small 10s of ppm variations in Li concentration is interesting but not a major factor in Li deposit formation, therefore this paper should not focus on Li deposits but discuss deep crustal Li variation in greater detail. The authors discuss later on in the paper that upper crustal differentiation plays a major role in the formation of Li deposits yet the paper does not discuss rhyolite data. There is also no mention of specific lithium deposits in the whole paper or the magmas/volcanic rocks associated with them, which I think is a critical piece to discussing Li deposits - especially exploration for them. E.g. Atacama, Hombre Muerto, Thacker Pass, Greenbushes. It is not sufficient to compile geochemical data and not relate them to specific lithium deposits (or even mention a Li deposit) in a paper about Li deposits.

We agree with the reviewer that we did not have that much to offer on the actual deposit formation. Our contribution primarily lies in the origin of the source rocks. We have refocused our paper to tackling the origin of Li rich source rocks, e.g., andesites and rhyolites. We do believe that the source rock is important, particularly rhyolitic tuffs, which can be enriched in Li significantly. Without this, it seems doubtful that basinal brines would even form.

The authors also need to do some more research on actual pegmatite deposits and how they form -there is a lot of data out there on the mechanisms involved in how they form. Even looking at the geological maps of the biggest hard rock resources in the world (Greenbushes, Pilgangoora, Tanco, etc.) one could see that the association with metapelites is ubiquitous. Extensive work has been done in this regard - there is a large volume of pegmatite literature out there (see London, among others).

We dropped pegmatite discussion, so no longer applicable.

I think there is interesting information in this paper but the authors need to rethink the purpose of the manuscript. I think it has a great contribution to understanding Li mobility in the deep crust but the extrapolation to Li resources is unresearched, unreferenced, and therefore unfounded.

We agree with the reviewer that we should rethink the purpose of the manuscript, which we have now done. We are thankful that the reviewer thinks our work on the geochemistry of Li in the crust is important. This is now the primary focus of the paper.

Line by line comments:

Line 15 (now in line 14). Nearly all lithium deposits are sourced from rhyolites and to a lesser extent, dacites, not andesites. In brine systems, 100% of producing brine deposits have rhyolitic tuff in the watershed. Salars without rhyolitic tuffs in the watershed do not have lithium deposits. In clay/sedimentary lithium deposits in the Basin and Range, lithium is sourced from rhyolite tuffs as well.

We have modified accordingly.

Line 17 (in the same line). Most brine deposits occur in back arc extensional basins and into intracontinental settings,

not in the continental arc itself. I don't see any discussion of any actual lithium deposits or associated magmas in this paper. I think this needs to be changed.

We have cut down the deposit section now.

Line 19 (now in line 15). Change to, "the development of Li-enriched brines is thus..."

Fixed, but abstract has been rewritten.

Line 36 (now in line 33). Should be, "Growth in DEMAND of hybrid..."

Fixed.

Line 40 (now in line 38). Li clay/sedimentary deposits are actually quite large. The Thacker Pass Project in NV is the 4th largest Li resource in the world. Oilfield brines and geothermal brines are inconsequential and the technology is not there to extract them yet.

Yes, we agree that the primary Li sources are from brines, pegmatites and, in some instances from the clays as compiled in our Supplementary Data. We here are trying to present an entire picture of Li sources including the minor oilfield brines and geothermal brines. Indeed, from the compilation Figure 1 and new compiled Supplementary Data, we can get a direct sense of which kind of Li ores are relatively world-widely distributed and crucial.

FIGURE 1A. This graph mainly just shows degree of erosion. Precambrian basement = old and therefore the pegmatites are exposed.

We agree. We have modified text so that we don't focus on the pegmatites. We only show the pegmatite ones here for completeness.

FIGURE 1B and 1C. These plots are extremely hard to read.

We have now adjusted the font and symbol size to make them more marked.

FIGURE 1D. Rhyolite tuff, not andesite tuff.

Fixed.

Line 76 (now removed). Partial melting of pelitic metamorphic rocks is a major source for pegmatites. Look at geological maps of the main deposits – the spodumene, Li-bearing pegmatites are nearly ubiquitously intruding these rocks.

We dropped pegmatite discussion, so no longer applicable.

FIGURE 2. Trends are there, but do not play a major role in what magma actually leads to a lithium deposit. A change in approximately 20 ppm Li in these magmas is inconsequential relative to what happens in magma chambers within the upper crust. The lithium concentration can vary 100s of ppm in a single rhyolitic magma chamber due to differentiation processes. That plays way more of a role than these few – 10s of ppm changes in the deep crust. It has long been known that Li is higher on continental crust.

We do not doubt that local enrichments within a given rhyolite can lead to very high Li concentrations. However, basinal Li deposits presumably derive their Li from surrounding watershed and thus the large scale Li concentration of the rocks in the watershed are of interest. Had these rocks been basaltic, scavenging from an even larger area would be needed. In any case, we have reduced our discussion of economic deposits, and focused more on the petrology and geochemistry.

Line 114-120 (now in line 108-114). This is interesting, but really just a commentary on the compatibility of Y. Li is increasing with SiO₂ content regardless of the Y.

Yes, we see how this was disjointed. We have now revised to make it flow more smoothly.

FIGURE 3. Is this plot necessary? This really is just showing that the degree of upper crustal differentiation is the main control on Li concentration, not slab differences.

Yes, we do think it is essential and hope to keep it, although we understand now why the reviewer asks the question. Note that this figure is not associated with the degree of any upper crustal differentiation. We only examine the primitive magmas here, so what we are looking at is primitive Li contents. As noted by other two reviewers, this paper has two main arguments. First, slab has no significant contribution to arc magma source. Second, continental arcs/thickened crust facilitate enrichment in Li relative to island arc/ thin crust. In Figure 3, we show that Li content in primitive arc magma correlates with crustal thickness but not slab thermal parameter, strongly suggesting crustal thickening, instead of subducting slab, results in greater Li enrichment in primitive magma of thickened arcs. Such an enrichment beneath thick crust will be delivered into further differentiation.

Line 163 (now in line 167). I think it is widely accepted that Li is coming from the crust, not the slab. Most Li deposits in the world occur from the back arc to intracontinental settings.

It is not widely accepted in the geochemical literature, but it is interesting that the economic geologists think it might come from the crust. Hopefully, our work can be used to show that the economic geologists are correct. That being said, the Li in the crust is ultimately from the mantle or the slab, and here what we are trying to show is that the slab is not important.

Line 164 (now in line 168). Once again, this is dealing in the 10s of ppms. It is inconsequential compared to processes in the upper crust which have a much greater control on Li enrichment. In a given tectonic setting, say in a back arc, if you have an andesite, there will be less than 50 ppm Li in the magma. If that same magma evolves, there could be 10x the amount of Li. Volcanic rocks that are sources for the main Li brine deposits in the Li Triangle have Li contents that vary by a factor of 5 within individual eruptions, consistent with fractional crystallization. The extent to which these magmas are within the watersheds of the salars is the controlling factor in the formation of Li deposits. The 10-20 ppm variations in arc magmas might play a tiny role very early on but it is inconsequential compared to this. Look at any non-producing salar in the Li triangle. None of them have dacite - rhyolite tuffs, most of them have andesites in their watersheds.

We have reduced our discussion on Li brines. We completely agree with the reviewer that differences between rhyolites are not important. The most important control on the brines themselves is the extent of the watershed, which may include many types of rhyolites. What we are trying to say, and have now clarified, is essentially why there are more rhyolites in continental arcs than in island arcs. And on top of this, rhyolites in continental arcs have more Li than rhyolites in thin arc settings. Had the watershed consisted of basalts or for that matter, rhyolites or andesites formed in thin arcs, a much larger watershed would be needed to scavenge more Li.

Line 184 (now in the line 190). Many Li deposits and Li-rich magmas are not associated with arc magmatism, though.

That is true. We have focused our manuscript on arc systems now.

Line 205 (now in line 206). This was the thesis of Benson et al. (2017)

Benson et al. 2017 argued that Li enrichment in some intracontinental magmas is due to assimilation of pre-existing felsic continental crust. We are arguing that crustal thickening facilitates both Li enrichment within primitive arc magma and subsequent differentiation, rather than by assimilation.

Line 209 (now in line 212). This is extensively studied and known. See work by Godfrey and Munk, among others. (Godfrey et al 2013), Munk et al (2016), Munk et al (2018). This work needs to be credited.

We have now added more relevant references.

Line 215 (now in line 217). These processes, documented in detail by other workers, are the main mechanisms leading to Li brine deposits – and really are disconnected from the data presented earlier.

We have focused more on the Li enrichment in magmatism.

Line 240 (now removed). The data is out there.

We deleted this section

Line 253 (now in line 231). This is not the main control.

We have reduced our discussion of economic deposits, and focused more on the Li enrichment in magmatism.

Reference:

- Brenan, J.M., Ryerson, F.J., Shaw, H.F., 1998. The role of aqueous fluids in the slab-to-mantle transfer of boron, beryllium, and lithium during subduction: experiments and models. *Geochim Cosmochim Acta* 62, 3337-3347.
- Dietrich, V., Emmermann, R., Oberhänsli, R., Puchelt, H., 1978. Geochemistry of basaltic and gabbroic rocks from the West Mariana Basin and the Mariana Trench. *Earth and Planetary Science Letters* 39, 127-144.
- Elliott, T., 2003. Tracers of the slab. *Geophysical Monograph-American Geophysical Union* 138, 23-46.
- Elliott, T., Jeffcoate, A., Bouman, C., 2004. The terrestrial Li isotope cycle: light-weight constraints on mantle convection. *Earth and Planetary Science Letters* 220, 231-245.
- Plank, T., 2014. The chemical composition of subducting sediments. *Treatise on geochemistry* 4, 607-629.
- Sharaskin, A.Y., 1982. PETROGRAPHY AND GEOCHEMISTRY OF BASEMENT ROCKS FROM 5 LEG-60 SITES. *Initial Reports of the Deep Sea Drilling Project* 60, 647-656.
- Tomascak, P.B., Widom, E., Benton, L.D., Goldstein, S.L., Ryan, J.G., 2002. The control of lithium budgets in island arcs. *Earth and Planetary Science Letters* 196, 227-238.

REVIEWERS' COMMENTS:

Reviewer #1 (Remarks to the Author):

Review NCOMMS-19-42377A (Revised manuscript)

The authors have addressed the issues I have raised in my initial review and they made adequate changes to the manuscript. I think the revised manuscript has improved significantly by focussing on Li in arc magmas and eliminating the discussion on pegmatites, which was somewhat "problematic" as pointed out in the reviews. The quantitative modelling that was added is a useful addition in support of the ideas presented. The major points of the manuscript are now much clearer and more convincing than in the initially submitted manuscript. Therefore, I only have a couple of minor comments to make:

103-105: Please check this statement: Li/Y shows only a subtle increase with increasing crustal thickness. This subtle correlation may result from melting residual garnet (or amphibole), which sequesters Y,

The formulation "melting a garnet-bearing residue" should be used, as "melting garnet" would increase Y in the melt and hence decrease Li/Y, whereas if garnet remains residual, Li/Y in the melt would increase. In the supplement, you explicitly mention "residual mantle garnet", which should be added here for clarity, too (you are not referring to garnet in a lower crustal magma source).

169-172: Absence of Li isotopic signatures in arc magmas: These new statements are interesting and independently supported by work on Li isotopes in veined mantle xenoliths. We investigated veined mantle xenoliths from Kamchatka that show MORB-source mantle-like $\delta^7\text{Li}$ values, providing additional support, independent from the arc magmas themselves, that Li from the slab does not appear to have a significant influence on the Li isotopic composition of melts that ascend through the sub-arc mantle. Interestingly, MORB-source mantle-like $\delta^7\text{Li}$ values were also found in arc lavas from Kamchatka in a very recent paper from April 2020. Both studies constitute a nice example in support of the hypothesis presented.

References:

Halama et al., 2009, *Contrib. Mineral. Petrol.* 158:197–222, DOI 10.1007/s00410-009-0378-5
Liu et al., 2020, *JGR Solid Earth*, <https://doi.org/10.1029/2019JB019237>

Review of revised manuscript by Ralf Halama (Keele University, UK), June 2020

Reviewer #2 (Remarks to the Author):

The authors made thoughtful and substantive changes to the original (strong) manuscript. I am satisfied with the revisions and think the manuscript is in good shape for publication (save for some minor edits, below).

line 16: "...landlocked..."

line 106: I think you mean "...arc magmas and crustal thickness or Li/Y..."

line 109: "...Li/Y remains constant."

line 115: "Previous studies suggested a slab origin based on high Li/Y in evolved arc magmatic rocks. These rocks had already experienced Li/Y fractionation, so the apparent slab signatures were false (Fig. 3b)."

line 162: "...Li is soluble..."

line 194: "...thick arcs demonstrate >10-fold enrichments..."

line 213: Maybe the more generic "volcanic/pyroclastic" instead of "tuffs" here?

line 214: Everything is weatherable. You made the point that they are easily weatherable in the previous sentence, so you can leave it out here.

line 215: Confusing of subject and object here: I think "where" should refer to "soluble Li," so it should read, "where it is further concentrated..."

line 231: "We have shown that..."

line 232: "Lithium enrichment..."

line 236: "...sources for the release of Li during weathering."

Reviewer #3 (Remarks to the Author):

- This paper argues that arcs and slab-derived fluids do not contribute to the Li content of magmas in continental arcs. It presents a great dataset of global arc magmas and does a great job of comparing the lithium content across a range of tectonic and slab settings.
- I think the paper falls short in describing the parts of the process that are most significant for generating high-lithium magmas, most notably the extent to which crustal assimilation plays a role in the differentiation process. This process is spoken about in generalities, and while I understand it is not the main focus of the paper, needs to be addressed in a more detailed paper. This is especially because the main conclusion of the paper is that the crustal processes are the most important parts of the Li enrichment.
- This conclusion, that thick continental crust is essential for generating high-Li magmas, is also one of the main conclusions of Benson et al. 2017 (Nat Comms). See figures 4a and 4b.
- Also, most large Li deposits and their source rocks (clay and brine) technically occur in back-arc settings where rhyolite is generated, not on the arc itself. It's a small technicality but I think it could be a significant one.
- One final point I would make is also discussed in my line items below, but I think it is critical because it is not discussed at all in the paper. the whole-rock Li content measured is not suitable for comparing magmatic Li contents due to (1) variation in alteration/weathering of volcanic rocks and the ease to which Li is mobilized during those processes; and (2) anywhere from 10-60+% of the Li in the original magma is lost during the eruption. It varies based on a wide variety of conditions. Therefore, pre-eruptive concentration of the magma can be very different from the post-eruptive rock measured - at levels significant enough to change the conclusions of this study. Therefore, I caution against using giant datasets on whole rock Li measurements to make generalizations on Li contents - especially because the Li is so low and minor amounts of weathering or differences in degassing during the eruptions can significantly change the Li content in the whole rock that is measured. For example, a certain tectonic setting could have magmas with lower volatile contents and therefore less partitioning of the Li into the gas phase leading to higher Li in the whole rock on the surface. This could be a regional phenomenon that plays a bigger role than anything presented in this paper. Just one example of a possible hypothesis that should be addressed - especially because the authors state the crustal differentiation plays the biggest role in the formation of Li deposits.

Line 10. Why does this lead with "Instead"?

Line 16. Land-locked

Lines 40-43. I would argue that the back arc is the location of much of the brines. (Olaroz, Cauchari, Uyuni, Hombre Muerto). Atacama is right on the arc and its source rocks are from the Pacana Caldera just to the east of the arc.

Line 81. I would add some citations here. This is a well-known concept.

Line 164. This is a great summary sentence.

Line 184. Is this due to fractionation or assimilation of continental crust?

Line 213. I don't know of a single lithium deposit that derived its Li from an andesite. I would exclude.

Line 219. What about assimilation? Are you arguing here that all of the Li comes from the mantle

wedge? That seems wrong based on what we know about the relative contributions of crust vs. arc material in arc and back arc settings.

Line 249. A big problem with using whole-rock Li concentrations is the degree of weathering and alteration. A rock sitting on the surface of the earth loses Li even with just passing groundwater (that's how brines form). In addition, any study that wants to compare Li concentrations of magmas cannot use whole-rock values (at the very least, needs to talk about the limitations thereof). One of the biggest concerns is loss of Li during the eruption to the vapor phase. As much as 60% of the Li is lost (Hofstra, Benson papers), and it varies based on the magmas. So, comparing whole rock data is like comparing apples and oranges. I understand data is limited but this issue, at the very least, needs to be addressed in the Supplemental Materials.

Revision notes

We thank again the three reviewers for their thorough reviews. We have made the suggested changes accordingly. We have also removed the "Conclusions" heading and done other style adjustments as the editor required. All changes in this manuscript are marked in red. Specific responses are below.

Reviewer #1 (Remarks to the Author):

Review NCOMMS-19-42377A (Revised manuscript)

The authors have addressed the issues I have raised in my initial review and they made adequate changes to the manuscript. I think the revised manuscript has improved significantly by focussing on Li in arc magmas and eliminating the discussion on pegmatites, which was somewhat "problematic" as pointed out in the reviews. The quantitative modelling that was added is a useful addition in support of the ideas presented. The major points of the manuscript are now much clearer and more convincing than in the initially submitted manuscript. Therefore, I only have a couple of minor comments to make:

103-105: Please check this statement: Li/Y shows only a subtle increase with increasing crustal thickness. This subtle correlation may result from melting residual garnet (or amphibole), which sequesters Y,

The formulation "melting a garnet-bearing residue" should be used, as "melting garnet" would increase Y in the melt and hence decrease Li/Y, whereas if garnet remains residual, Li/Y in the melt would increase. In the supplement, you explicitly mention "residual mantle garnet", which should be added here for clarity, too (you are not referring to garnet in a lower crustal magma source).

Clarified.

169-172: Absence of Li isotopic signatures in arc magmas: These new statements are interesting and independently supported by work on Li isotopes in veined mantle xenoliths. We investigated veined mantle xenoliths from Kamchatka that show MORB-source mantle-like $\delta^7\text{Li}$ values, providing additional support, independent from the arc magmas themselves, that Li from the slab does not appear to have a significant influence on the Li isotopic composition of melts that ascend through the sub-arc mantle. Interestingly, MORB-source mantle-like $\delta^7\text{Li}$ values were also found in arc lavas from Kamchatka in a very recent paper from April 2020. Both studies constitute a nice example in support of the hypothesis presented.

References:

Halama et al., 2009, *Contrib. Mineral. Petrol.* 158:197–222, DOI 10.1007/s00410-009-0378-5
Liu et al., 2020, *JGR Solid Earth*, <https://doi.org/10.1029/2019JB019237>

Additional references added.

Review of revised manuscript by Ralf Halama (Keele University, UK), June 2020

Reviewer #2 (Remarks to the Author):

The authors made thoughtful and substantive changes to the original (strong) manuscript. I am satisfied with the revisions and think the manuscript is in good shape for publication (save for some minor edits, below).

line 16: "...landlocked..."

line 106: I think you mean "...arc magmas and crustal thickness or Li/Y..."

line 109: "...Li/Y remains constant."

line 115: "Previous studies suggested a slab origin based on high Li/Y in evolved arc magmatic rocks. These rocks had already experienced Li/Y fractionation, so the apparent slab signatures were false (Fig.

3b)."

line 162: "...Li is soluble..."

line 194: "...thick arcs demonstrate >10-fold enrichments..."

line 213: Maybe the more generic "volcanic/pyroclastic" instead of "tuffs" here?

line 214: Everything is weatherable. You made the point that they are easily weatherable in the previous sentence, so you can leave it out here.

line 215: Confusing of subject and object here: I think "where" should refer to "soluble Li," so it should read, "where it is further concentrated..."

line 231: "We have shown that..."

line 232: "Lithium enrichment..."

line 236: "...sources for the release of Li during weathering."

All edits pointed out above have been adjusted accordingly.

Reviewer #3 (Remarks to the Author):

- This paper argues that arcs and slab-derived fluids do not contribute to the Li content of magmas in continental arcs. It presents a great dataset of global arc magmas and does a great job of comparing the lithium content across a range of tectonic and slab settings.
- I think the paper falls short in describing the parts of the process that are most significant for generating high-lithium magmas, most notably the extent to which crustal assimilation plays a role in the differentiation process. This process is spoken about in generalities, and while I understand it is not the main focus of the paper, needs to be addressed in a more detailed paper. This is especially because the main conclusion of the paper is that the crustal processes are the most important parts of the Li enrichment.

Crustal assimilation is unlikely to be the dominant process. However, we have added a sentence in the paper highlighting that crustal assimilation could be superimposed on the differentiation processes we describe.

- This conclusion, that thick continental crust is essential for generating high-Li magmas, is also one of the main conclusions of Benson et al. 2017 (Nat Comms). See figures 4a and 4b.

We have modified our wording to give credit to this.

- Also, most large Li deposits and their source rocks (clay and brine) technically occur in back-arc settings where rhyolite is generated, not on the arc itself. It's a small technicality but I think it could be a significant one.

The reviewer is correct if what is meant by "back-arc" is behind the main arc front, but still within the arc orogen. We have added a sentence to this effect and have added additional citations.

- One final point I would make is also discussed in my line items below, but I think it is critical because it is not discussed at all in the paper. the whole-rock Li content measured is not suitable for comparing magmatic Li contents due to (1) variation in alteration/weathering of volcanic rocks and the ease to which Li is mobilized during those processes; and (2) anywhere from 10-60+% of the Li in the original magma is lost during the eruption. It varies based on a wide variety of conditions. Therefore, pre-eruptive concentration of the magma can be very different from the post-eruptive rock measured - at levels significant enough to change the conclusions of this study. Therefore, I caution against using giant datasets on whole rock Li measurements to make generalizations on Li contents - especially because the Li is so low and minor amounts of weathering or differences in degassing during the eruptions can significantly change the Li content in the whole rock that is measured. For example, a certain tectonic setting could have magmas with lower volatile contents and therefore less partitioning of the Li into the gas phase leading to higher Li in the whole rock on the surface. This could be a regional phenomenon that plays a bigger role than anything presented in this paper. Just one example of a possible hypothesis that should be addressed - especially because the authors state

the crustal differentiation plays the biggest role in the formation of Li deposits.

We have added a caveat sentence to the manuscript, recognizing that degassing and weathering could compromise whole rock compositions. We note in this caveat that these effects will be most pronounced for subaerial volcanoes and so will affect evolved magmas the most. Thus, our enrichment factors from whole rocks are minimum bounds.

In any case, the Li contents of whole rocks probably are representative of Li in the magma, suggesting these degassing or alteration effects, while they do happen, probably are not important enough to change global systematics

- Firstly, in our data filtering, we excluded the samples of the oxide sum beyond the range of 98-101.5 wt.% to filter out the altered samples;
- It is well documented that global arc lavas (whole rocks) have mantle-like Li isotopic signatures (Halama et al., 2009; Liu et al., 2020; Tomascak et al., 2000; Tomascak et al., 2002). Given that alteration or degassing will significantly modify the Li isotopic compositions, the mantle-like Li isotopic value indicates that degassing has negligible impacts on Li.
- As shown below (unpublished results) and Fig.2b in manuscript, the whole rock Li contents fall on a continuous trend with the very Li-rich melt inclusions (e.g. Benson et al.). If the melt inclusions represent pristine unaltered Li, then the fact that whole rocks fall along an array continuous with the melt inclusions suggests that they have not been significantly disturbed. Moreover, the entire array (melt inclusions and whole rocks) can be explained by a 1/F relationship.

Fig.R1 Large dots denote the average of Li concentrations binned by relative residual melt fractions, F, in 0.02 increments of F. F is calculated as initial potassium oxide contents of primary arc magma, K_2O_0 , over those of evolved melts, K_2O_1 . Colored lines represent modelling results based on Rayleigh fractionation with specific conditions.

Line 10. Why does this lead with "Instead"?

We would prefer to keep this because it emphasizes that our findings go against the wisdom illustrated in the previous sentence.

Line 16. Land-locked

Fixed.

Lines 40-43. I would argue that the back arc is the location of much of the brines. (Olaroz, Cauchari, Uyuni, Hombre Muerto). Atacama is right on the arc and its source rocks are from the Pacana Caldera just to the east of the arc.

That is correct. We have corrected accordingly.

Line 81. I would add some citations here. This is a well-known concept.

Our data show Li is not that moderately incompatible. It is highly to moderately incompatible. Citations added.

Line 164. This is a great summary sentence.

Thank you. It is a new conclusion here.

Line 184. Is this due to fractionation or assimilation of continental crust?

It is due to crystal fractionation as demonstrated by the data and modeling in manuscript (see Fig. 2).

Line 213. I don't know of a single lithium deposit that derived its Li from an andesite. I would exclude.

All surficial weatherable rocks contribute Li, but we have added sentences to emphasize the importance of rhyolite.

Line 219. What about assimilation? Are you arguing here that all of the Li comes from the mantle wedge? That seems wrong based on what we know about the relative contributions of crust vs. arc material in arc and back arc settings.

We may not be able to solve all the issues in arc science... I think a lot of the community is now moving towards mantle derived rather than crustal assimilation.

Line 249. A big problem with using whole-rock Li concentrations is the degree of weathering and alteration. A rock sitting on the surface of the earth loses Li even with just passing groundwater (that's how brines form). In addition, any study that wants to compare Li concentrations of magmas cannot use whole-rock values (at the very least, needs to talk about the limitations thereof). One of the biggest concerns is loss of Li during the eruption to the vapor phase. As much as 60% of the Li is lost (Hofstra, Benson papers), and it varies based on the magmas. So, comparing whole rock data is like comparing apples and oranges. I understand data is limited but this issue, at the very least, needs to be addressed in the Supplemental Materials.

...We have explained in the above.

Reference:

Benson, T.R., Coble, M.A., Rytuba, J.J., Mahood, G.A., 2017. Lithium enrichment in intracontinental rhyolite magmas leads to Li deposits in caldera basins. *Nat Commun* 8, 270.

Halama, R., Savov, I.P., Rudnick, R.L., McDonough, W.F., 2009. Insights into Li and Li isotope cycling and sub-arc metasomatism from veined mantle xenoliths, Kamchatka. *Contrib Mineral Petr* 158, 197-222.

Liu, H., Xiao, Y., Sun, H., Tong, F., Heuser, A., Churikova, T., Wörner, G., 2020. Trace Elements and Li Isotope Compositions Across the Kamchatka Arc: Constraints on Slab-Derived Fluid Sources. *Journal of Geophysical Research: Solid Earth* 125, e2019JB019237.

Tomascak, P.B., Ryan, J.G., Defant, M.J., 2000. Lithium isotope evidence for light element decoupling in the Panama subarc mantle. *Geology* 28, 507-510.

Tomascak, P.B., Widom, E., Benton, L.D., Goldstein, S.L., Ryan, J.G., 2002. The control of lithium budgets in island arcs. *Earth and Planetary Science Letters* 196, 227-238.